# Adaptive Multi-model Fusion Learning for Sparse-reward Reinforcement Learning

## Abstract

In this paper, we consider intrinsic reward generation for sparse-reward reinforcement learning based on model prediction errors. In typical model-prediction-error-based intrinsic reward generation, an agent has a learning model for the underlying environment. Then, intrinsic reward is designed as the error between the model prediction and the actual outcome of the environment, based on the fact that for less-visited or non-visited states, the learned model yields larger prediction errors, promoting exploration helpful for reinforcement learning. This paper generalizes this model-prediction-error-based intrinsic reward generation method to multiple prediction models. We propose a new adaptive fusion method relevant to the multiple-model case, which learns optimal prediction-error fusion across the learning phase to enhance the overall learning performance. Numerical results show that for representative locomotion tasks, the proposed intrinsic reward generation method outperforms most of the previous methods, and the gain is significant in some tasks.

## 1 Introduction

Reinforcement learning (RL) with sparse reward is an active research area (Andrychowicz et al., 2017; Tang et al., 2017; de Abril & Kanai, 2018; Oh et al., 2018; Kim et al., 2019). In sparse-reward RL, the environment does not return a non-zero reward for every agent's action but returns a non-zero reward only when certain conditions are met. Such situations are encountered in many action control problems (Houthooft et al., 2016; Andrychowicz et al., 2017; Oh et al., 2018). As in conventional RL, exploration is essential at the early stage of learning in sparse-reward RL, whereas the balance between exploration and exploitation is required later.

Intrinsically motivated RL has been studied to stimulate better exploration by generating intrinsic reward for each action by the agent itself. Recently, many intrinsically-motivated RL algorithms have been devised especially to deal with the sparsity of reward, e.g., based on the notion of curiosity (Houthooft et al., 2016; Pathak et al., 2017), surprise (Achiam & Sastry, 2017). In essence, in these intrinsic reward generation methods, the agent has a learning model for the next state or the transition probability of the underlying environment, and intrinsic reward is designed as the error between the model prediction and the actual outcome of the environment, based on the fact that for less-visited or non-visited states, the learned model yields larger prediction errors, promoting exploration helpful for reinforcement learning. These previous methods typically use a single prediction model for the next state or the environment's transition probability.

In this paper, we generalize this model-prediction-error-based approach to the case of multiple prediction models and propose a new framework for intrinsic reward generation based on the optimal adaptive fusion of multiple values from multiple models. The use of multiple models increases diversity in modeling error values and the chance to design a better intrinsic reward from these values. The critical task is to learn an optimal fusion rule to maximize the performance across the entire learning phase. In order to devise such an optimal adaptive fusion algorithm, we adopt the $\alpha$-mean with the scale-free property from the field of information geometry (Amari, 2016) and apply the meta-gradient optimization to search for optimal fusion at each stage of learning. Numerical results show that the proposed multi-model intrinsic reward generation combined with fusion learning significantly outperforms existing intrinsic reward generation methods.

## 2 RELATED WORK

Intrinsically-motivated RL and exploration methods can be classified mainly into two categories. One is to explicitly generate intrinsic reward and train the agent with the sum of the extrinsic reward and the adequately scaled intrinsic reward. The other is indirect methods that do not explicitly generate intrinsic reward. Our work belongs to the first category, and we conducted experiments using baselines in the first category. However, we also detailed the second category in Appendix H for readers for further work in the intrinsically-motivated RL area.

Houthooft et al. (2016) used the information gain on the prediction model as an additional reward based on the notion of curiosity. Tang et al. (2017) efficiently applied count-based exploration to high-dimensional state space by mapping the states' trained features into a hash table. The concept of surprise was exploited to yield intrinsic rewards (Achiam & Sastry, 2017). Pathak et al. (2017) defined an intrinsic reward with the prediction error using a feature state space, and de Abril & Kanai (2018) enhanced Pathak et al. (2017)'s work with the idea of homeostasis in biology.

Zheng et al. (2018) used a delayed reward environment to propose training the module to generate intrinsic reward apart from training the policy. This delayed reward environment for sparse-reward settings differs from the previous sparse-reward environment based on thresholding (Houthooft et al., 2016). (The agent gets a non-zero reward when the agent achieves a specific physical quantity - such as the distance from the origin - larger than the predefined threshold.) Pathak et al. (2019) interpreted the disagreement among the models as the variance of the predicted next states and used the variance as the final differentiable intrinsic reward. Our method is a generalized version of their work as we can apply our proposed fusion method to the multiple squared error values between a predicted next state and all the predicted next states' average. Freirich et al. (2019) proposed generating intrinsic reward by applying a generative model with the Wasserstein-1 distance. With the concept of state-action embedding, Kim et al. (2019) adopted the Jensen-Shannon divergence (JSD) (Hjelm et al., 2019) to construct a new variational lower bound of the corresponding mutual information, guaranteeing numerical stability. Our work differs from these two works in that we use the adaptive fusion method of multiple intrinsic reward at every timestep.

## 3 THE PROPOSED METHOD

### 3.1 SETUP

We consider a discrete-time continuous-state Markov Decision Process (MDP), denoted as $(\mathcal{S}, \mathcal{A}, P, r, \rho_0, \gamma)$, where $\mathcal{S}$ and $\mathcal{A}$ are the sets of states and actions, respectively, $P : \mathcal{S} \times \mathcal{A} \to \Pi(\mathcal{S})$ is the transition probability function, where $\Pi(\mathcal{S})$ is the space of probability distributions over $\mathcal{S}$, $r : \mathcal{S} \times \mathcal{A} \times \mathcal{S} \to \mathbb{R}$ is the *extrinsic* reward function, $\rho_0$ is the probability distribution of the initial state, and $\gamma$ is the discounting factor. A (stochastic) policy is represented by $\pi : \mathcal{S} \to \Pi(\mathcal{A})$, where $\Pi(\mathcal{A})$ is the space of probability distributions on $\mathcal{A}$ and $\pi(a|s)$ represents the probability of choosing action $a \in \mathcal{A}$ for given state $s \in \mathcal{S}$. In sparse-reward RL, the environment does not return a non-zero reward for every action but returns a non-zero reward only when certain conditions are met by the current state, the action and the next state (Houthooft et al., 2016; Andrychowicz et al., 2017; Oh et al., 2018). Our goal is to optimize the policy $\pi$ to maximize the expected cumulative return $\eta(\pi)$ by properly generating intrinsic reward in such sparse-reward environments. We assume that the true transition probability distribution $P$ is unknown to the agent.

### 3.2 INTRINSIC REWARD DESIGN BASED ON MODEL PREDICTION ERRORS

Intrinsically-motivated RL adds a properly designed intrinsic reward at every timestep $t$ to the actual extrinsic reward to yield a non-zero total reward for training even when the extrinsic reward returned by the environment is zero (Pathak et al., 2017; Tang et al., 2017; de Abril & Kanai, 2018). In the model-prediction-error-based intrinsic reward design, the agent has a prediction model parametrized by $\phi$ for the next state $s_{t+1}$ or the transition probability $P(s_{t+1}|s_t, a_t)$, and the intrinsic reward is designed as the error between the model prediction and the actual outcome of the environment (Houthooft et al., 2016; Achiam & Sastry, 2017; Pathak et al., 2017; Burda et al., 2019; de Abril & Kanai, 2018). Thus, the intrinsic-reward-incorporated problem under this approach is given in most

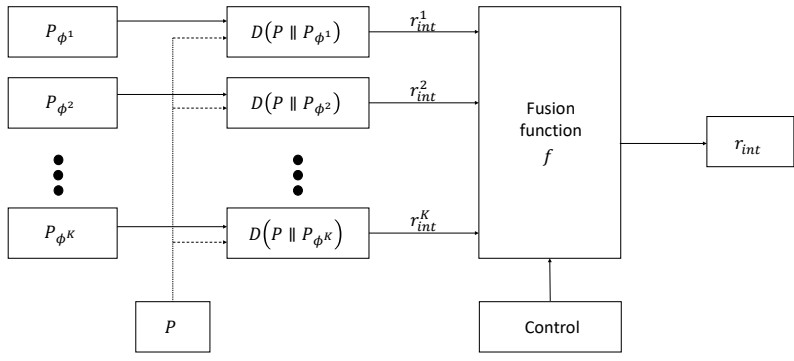

Figure 1: Adaptive fusion of $K$ prediction errors from the multiple models

cases as

$$\max_{\pi} \left\{ \eta(\pi) + c \, \mathbb{E}_{(s,a) \sim \pi} [D(P||P_\phi)|(s,a)] \right\} \tag{1}$$

for some constant $c > 0$ and some divergence function $D(\cdot||\cdot)$, where $\eta(\pi)$ is the cumulative reward associated with policy $\pi$, and $P_\phi$ is the learning model parameterized by $\phi$ that the agent has regarding the true unknown transition probability $P$ of the environment. For the divergence, the mean squared error (MSE) between the actual next state and the predicted next state can be used for the error measure when the learning model predicts the next state itself, or alternatively the Kullback-Leibler divergence (KLD) between the probability distribution for the next state $s_{t+1}$ and the predicted probability distribution for $s_{t+1}$ can be used when the learning models learn the transition probability. In the case of KLD, the intractable $D_{KL}(P||P_\phi)|(s,a)$ with unknown $P$ can be approximated based on the 1-step approximation (Achiam & Sastry, 2017).

### 3.3 THE PROPOSED ADAPTIVE FUSION LEARNING

We consider using multiple prediction models and the design of prediction-error-based intrinsic reward from the multiple models. Suppose we have a collection of $K(\geq 2)$ models parametrized by $\phi^1, \cdots, \phi^K$ to generate $K$ prediction error (approximation) values at timestep $t$ as intrinsic reward $r_{t,int}^j(s_t, a_t, s_{t+1}), j = 1, \cdots, K$, respectively. The key problem of multi-model prediction-error-based intrinsic reward design is how to learn $\phi^1, \cdots, \phi^K$ and how to optimally fuse the $K$ values $r_{t,int}^j(s_t, a_t, s_{t+1}), j = 1, \cdots, K$, to generate a single intrinsic reward to be added to the scalar cumulative return for policy update. The considered multi-model fusion structure is shown in Fig. 1. To fuse the $K$ values for a single reward value, one can use one of the known methods such as average, minimum, or maximum. However, there is no guarantee of optimality for such arbitrary choices, and one fixed fusion rule may not be optimal for the entire learning phase.

Let a fusion function be denoted as

$$r_{int} = f(r_{int}^1, r_{int}^2, \cdots, r_{int}^K), \tag{2}$$

where $r_{int}^1, r_{int}^2, \cdots, r_{int}^K$ are the $K$ input values and $r_{int}$ is the output value. To devise an optimal adaptive fusion rule, we consider the following requirements for the fusion function $f$.

**Condition 1.** *The fusion function $f$ varies with some control parameter to adapt to the relative importance of the $K$ input values.*

We require Condition 1 so that the fusion of the $K$ input values can adapt to the learning situation. When the more aggressive fusion is required at some phase of learning, we want the function $f$ to be more like maximum. On the other hand, when the more conservative fusion is required at other learning phases, we want the function $f$ to be more like minimum. Furthermore, we want this optimal adaptation is learned based on data to yield maximum cumulative return. In addition, we impose the following relevant condition for any reasonable fusion function:

**Condition 2.** *The fusion function $f$ is scale-free, i.e.,*

$$f(cr_{int}^1, cr_{int}^2, \cdots, cr_{int}^K) = cf(r_{int}^1, r_{int}^2, \cdots, r_{int}^K). \tag{3}$$

Condition 2 implies that when we scale all the input values by the same factor $c$, the output is the $c$-scaled version of the fusion output of the not-scaled inputs.

Condition 2 is a proper requirement for any reasonable averaging function. The necessity of Condition 2 is explained in detail in Appendix G. Such a fusion function can be found based on the $\alpha$-mean of positive measures in the field of information geometry (Amari, 2016). For any $K$ positive[1] values $x_1, \cdots, x_K > 0$, the $\alpha$-mean of $x_1, \cdots, x_K$ is defined as

$$f_\alpha(x_1, \cdots, x_K) = h^{-1}\left(\frac{1}{K}\sum_{i=1}^{K} h(x_i)\right) \tag{4}$$

where $h(x)$ is given by the $\alpha$-embedding transformation:

$$h(x) = \begin{cases} x^{\frac{1-\alpha}{2}}, & \text{if } \alpha \neq 1 \\ \log x, & \text{if } \alpha = 1 \end{cases}. \tag{5}$$

It is proven that the unique class of transformation $h$ satisfying Condition 2 under the twice-differentiability and the strict monotonicity of $h$ is given by the $\alpha$-embedding (5) (Amari, 2007; 2016). Basically, Condition 2 is used to write $f_\alpha(cx_1, \cdots, cx_K) = h^{-1}\left(\frac{1}{K}\sum_{i=1}^{K} h(cx_i)\right) = cf_\alpha(x_1, \cdots, x_K)$. Taking $h(\cdot)$ on both sides yields $h(cf_\alpha(x_1, \cdots, x_K)) = \frac{1}{K}\sum_{i=1}^{K} h(cx_i)$. Then, taking partial derivative with respect to $x_i$ ($1 \leq i \leq K$) on both sides, we can show that the equation (5) is the unique class of mapping functions (Amari, 2007; 2016).

Furthermore, by varying $\alpha$, the $\alpha$-mean includes all numeric fusions with the scale-free property such as minimum, maximum, and conventional mean functions (Amari, 2016). When $\alpha = -\infty$, $f_\alpha(x_1, \cdots, x_K) = \max_i x_i$. On the other hand, when $\alpha = \infty$, $f_\alpha(x_1, \cdots, x_K) = \min_i x_i$. As $\alpha$ increases from $-\infty$ to $\infty$, the $\alpha$-mean output varies monotonically from maximum to minimum. See Appendix B. Hence, we can perform aggressive fusion to conservative fusion by controlling the parameter $\alpha$.

### 3.3.1 LEARNING OF $\alpha$ WITH META-GRADIENT OPTIMIZATION

In the proposed adaptive fusion, we need to adaptively control $\alpha$ judiciously to maximize the expected cumulative extrinsic return $\eta(\pi)$. To learn optimal $\alpha$ maximizing $\eta(\pi)$, we use the meta gradient method (Xu et al., 2018; Zheng et al., 2018). Optimal $\alpha$ at each stage of learning is learned with the proposed method, and it will be shown that optimal $\alpha$ varies according to the stage of learning. For policy $\pi_\theta$ with policy parameter $\theta$, let us define the following quantities.

• $\eta(\pi_\theta) = \mathbb{E}_{\tau \sim \pi_\theta}\left[\sum_{t=0}^{\infty} \gamma^t r(s_t, a_t, s_{t+1})\right]$: the expected cumulative sum of extrinsic rewards which we want to maximize. Here, $\tau$ is a sample trajectory.

• $\eta_{\text{total}}(\pi_\theta) = \mathbb{E}_{\tau \sim \pi_\theta}\left[\sum_{t=0}^{\infty} \gamma^t (r(s_t, a_t, s_{t+1}) + cf_\alpha(s_t, a_t, s_{t+1}))\right]$: the expected cumulative sum of both extrinsic and intrinsic rewards with which the policy $\pi_\theta$ is updated. Here, the dependence of the fusion output $f_\alpha$ on $(s_t, a_t, s_{t+1})$ through $r_{t,int}^j(s_t, a_t, s_{t+1})$ is shown with notation simplification.

Then, for a given trajectory $\tau = (s_0, a_0, s_1, a_1, \ldots)$ generated by $\pi_\theta$, we update $\theta$ towards the direction of maximizing $\eta_{total}(\pi_\theta)$:

$$\tilde{\theta} = \theta + \delta_\theta \nabla_\theta \eta_{total}(\pi_\theta) \tag{6}$$

where $\delta_\theta$ is the learning rate for $\theta$. Then, the fusion parameter $\alpha$ is updated to maximize the expected cumulated sum of extrinsic rewards for the updated policy $\pi_{\tilde{\theta}}$:

$$\tilde{\alpha} = \alpha + \delta_\alpha \nabla_\alpha \eta(\pi_{\tilde{\theta}}) \tag{7}$$

---

[1]When an input value to the $\alpha$-mean is negative due to divergence approximation in some cases, we can use exponentiation at the input stage and its inverse logarithm at the output stage. We used the exponentiation $\exp(-x)$ at the input stage with input $x$ and the negative logarithm of the $\alpha$-mean as its inverse at the output stage for actual implementation. In this case, due to the monotone decreasing property of the input mapping: $x \to \exp(-x)$, the output is the maximum when $\alpha = \infty$ and is the minimum when $\alpha = -\infty$.

where $\delta_\alpha$ is the learning rate for $\alpha$. Note that we update the policy parameter $\theta$ to maximize $\eta_{\text{total}}(\pi_\theta)$ so that the updated policy parameter $\tilde{\theta}$ is a function of $\alpha$. Therefore, $\nabla_\alpha \eta(\pi_{\tilde{\theta}})$ is not zero and can be computed by chain rule:

$$\nabla_\alpha \eta(\pi_{\tilde{\theta}}) = \nabla_{\tilde{\theta}} \eta(\pi_{\tilde{\theta}}) \, \nabla_\alpha \tilde{\theta} \qquad (8)$$

To learn optimal $\alpha$ together with $\theta$, we adopt an alternating optimization method widely used in meta-parameter optimization. That is, we iterate the following two steps in an alternating manner:

1) Update the policy parameter $\theta$ to maximize $\eta_{\text{total}}(\pi_\theta)$.

2) Update the fusion parameter $\alpha$ to maximize $\eta(\pi_{\tilde{\theta}})$, where $\tilde{\theta}$ is the updated policy parameter from Step 1.

In this way, we can learn proper $\alpha$ adaptively over timesteps to maximize the performance.

### 3.4 IMPLEMENTATION

We consider the case of $D(\cdot||\cdot) = D_{KL}(\cdot||\cdot)$ for implementation example (See Appendix F for the comparison of KLD and MSE). We use a collection of $K$ prediction models $P_{\phi^1}, \cdots, P_{\phi^K}$. Then, from the $j$-th model $P_{\phi^j}$, $j = 1, \cdots, K$, we have the $j$-th prediction error, given by

$$D_{KL}(P||P_{\phi^j})|(s_t, a_t) = \mathbb{E}_P \left[ \log \frac{P(\cdot|s_t, a_t)}{P'(\cdot|s_t, a_t)} \frac{P'(\cdot|s_t, a_t)}{P_{\phi^j}(\cdot|s_t, a_t)} \right] \geq \mathbb{E}_P \left[ \log \frac{P'(\cdot|s_t, a_t)}{P_{\phi^j}(\cdot|s_t, a_t)} \right]. \quad (9)$$

Note that the $j$-th model prediction error $D_{KL}(P||P_{\phi^j})|(s_t, a_t)$ is lower bounded as (9) for any distribution $P'$. In order to obtain a tight lower bound, $P'$ should be learned to be close to the true transition probability $P$. For increased degrees of freedom for better learning and estimation, we use the mixture distribution of $P_K = \sum_{i=1}^K q_i P_{\phi^i}$ for $P'$ with the learnable mixing coefficients $q_i \geq 0$ and $\sum_{i=1}^K q_i = 1$. The mixture model $P_K$ has increased model order for modeling the true $P$ beyond single-mode distribution. Then, the prediction error approximation as intrinsic reward for the $j$-th model $P_{\phi^j}$ at timestep $t$ is determined as $r_{t,int}^j(s_t, a_t, s_{t+1}) = \log \frac{P_K(s_{t+1}|s_t, a_t)}{P_{\phi^j}(s_{t+1}|s_t, a_t)}$, $j = 1, \cdots, K$. Note that each $r_{t,int}^j$ can be negative although the KLD is always nonnegative.

Although the proposed intrinsic reward generation method can be combined with general RL algorithms, we consider the PPO algorithm (Schulman et al., 2017), a popular on-policy algorithm generating a batch of experiences of length $L$ with every current policy. Thus, the exposition below is focused on application to PPO. For the $K$ prediction models $P_{\phi^1}, \cdots, P_{\phi^K}$, we adopt the fully-factorized Gaussian distribution (Houthooft et al., 2016; Achiam & Sastry, 2017). Then, $P_K$ becomes the class of $K$-modal Gaussian mixture distributions.

We first update the prediction models $P_{\phi^1}, \cdots, P_{\phi^K}$ and the corresponding mixing coefficients $q_1, \ldots, q_K$. In the beginning, the parameters $\phi^1, \cdots, \phi^K$ are independently initialized, and $q_i$'s are set to $\frac{1}{K}$ for all $i = 1, \cdots, K$. At every batch period $l$ of PPO, to jointly learn $\phi^i$ and $q_i$, we apply maximum-likelihood estimation (MLE) with an $L_2$-norm regularizer with KL constraints (Williams & Rasmussen, 2006; Achiam & Sastry, 2017):

$$\underset{\phi^i \, q_i, \, 1 \leq i \leq K}{\text{maximize}} \quad \underbrace{\mathbb{E}_{(s,a,s')} \log \left\{ \sum_{i=1}^K q_i P_{\phi^i}(s'|s,a) \right\}}_{=:\mathcal{L}_{\text{likelihood}}} - c_{\text{reg}} \underbrace{\sum_{i=1}^K \|\phi^i\|^2}_{=:\mathcal{L}_{reg}} \qquad (10)$$

$$\text{subject to} \quad \mathbb{E}_{(s,a)} \left[ D_{KL}(P_{\phi^i}||P_{\phi_{\text{old}}^i})(s,a) \right] \leq \kappa, \, \sum_{i=1}^K q_i = 1$$

where $\phi_{\text{old}}^i$ is the parameter of the $i$-th model before the update caused by (10), $c_{\text{reg}}$ is the regularization coefficient, and $\kappa$ is a positive constant. To solve this optimization problem with respect to $\{\phi^i\}$, we apply the method based on second-order approximation (Schulman et al., 2015a). For the update of $\{q_i\}$, we apply the EM method proposed in Dempster et al. (1977) and set $q_i$ as

$$q_i = \mathbb{E}_{(s,a,s')} \frac{q_i^{\text{old}} P_{\phi^i}(s'|s,a)}{\sum_{j=1}^K q_j^{\text{old}} P_{\phi^j}(s'|s,a)} \quad (1 \leq i \leq K) \qquad (11)$$

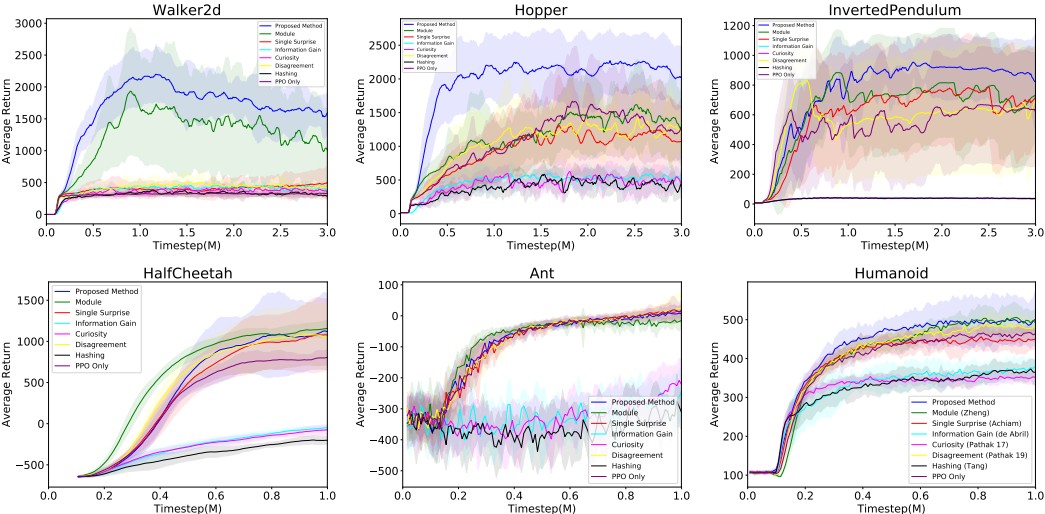

Figure 2: **Performance comparison.** All simulations were conducted over ten fixed random seeds. The $y$-axis in each figure with the title "Average Return" represents the mean value of the extrinsic returns of the most recent 100 episodes averaged over the ten random seeds. Each colored band in every figure represents the interval of $\pm\sigma$ around the mean curve, where $\sigma$ is the standard deviation of the ten instances of data from the ten random seeds. In order to give sufficient time steps for each environment, for the three environments in the top row, the experiments were performed for 3M timesteps. For the environments in the bottom row, the experiments were conducted for 1M timesteps. (For clarity, the first author of each of the algorithms is shown in the Humanoid plot.)

where $q_i^{\text{old}}$ is the mixing coefficient of the $i$-th model before the update caused by (11). For numerical stability, we use the "log-sum-exp" trick for computing (11) as well as $\mathcal{L}_{\text{likelihood}}$ defined in (10) and $\nabla_{\phi^i}\mathcal{L}_{\text{likelihood}}$. In addition, we apply simultaneous update of all $\phi^i$'s and $q_i$'s, which was found to perform better than one-by-one alternating update of the $K$ models for the considered case.

The update of policy by using PPO is as follows. Let $D$ be the batch of experiences for training the policy, i.e., $D = (s_t, a_t, r_t^{total}, s_{t+1}, \cdots, r_{t+L-2}^{total}, s_{t+L-1}, a_{t+L-1}, r_{t+L-1}^{total})$, where $a_t \sim \pi_{\theta_l}(\cdot|s_t)$, $s_{t+1} \sim P(\cdot|s_t, a_t)$, and $r_t^{total}$ is the total reward described below. Here, $\pi_{\theta_l}$ is the parameterized policy at the batch period $l$ corresponding to timestep $t, \cdots, t+L-1$ (the batch period index $l$ is included in $\pi_{\theta_l}$ for clarity). The total reward at timestep $t$ for training the policy is given by

$$r_t^{total}(s_t, a_t, s_{t+1}) = r_t(s_t, a_t, s_{t+1}) + \beta r_{t,int}(s_t, a_t, s_{t+1}) \tag{12}$$

where $r_t(s_t, a_t, s_{t+1})$ is the actual sparse extrinsic reward at timestep $t$ from the environment, $r_{t,int}(s_t, a_t, s_{t+1})$ is the intrinsic reward at timestep $t$, and $\beta > 0$ is the weighting factor. Here, for actual computation of the intrinsic reward, we further applied two techniques: the 1-step technique and the normalization technique used in Achiam & Sastry (2017) (which are described in Appendix C). Then, the policy $\pi_{\theta_l}$ is updated at every batch period $l$ with $D$ by following the standard PPO procedure based on the total reward (12). Summarizing the above, we provide the pseudocode of our algorithm, Algorithm 1, which assumes PPO as the base algorithm, in Appendix A.

# 4 RESULTS

## 4.1 PERFORMANCE COMPARISON

To evaluate the performance, we considered sparse-reward environments for continuous control. The considered tasks were six environments of Mujoco (Todorov et al., 2012), OpenAI Gym (Brockman et al., 2016): Walker2d, Hopper, InvertedPendulum, HalfCheetah, Ant, and Humanoid. To implement a sparse-reward setting, we adopted the delay method (Oh et al., 2018). We first accumulate extrinsic rewards generated from the considered environments for every $\Delta$ timesteps or until the episode ends.

Then we provide the accumulated sum of rewards to the agent at the end of the $\Delta$ timesteps or at the end of the episode, and repeat this process. For our experiments, we set $\Delta = 40$ as used in (Zheng et al., 2018). We compared the proposed method with existing intrinsic reward generation methods by using PPO as the base algorithm. We considered the existing intrinsic reward generation methods: single-model surprise (Achiam & Sastry, 2017), curiosity (Pathak et al., 2017), hashing (Tang et al., 2017), and information gain approximation (de Abril & Kanai, 2018). We also considered the method using intrinsic reward module (Zheng et al., 2018) among the most recent works introduced in Section 2, which uses delayed sparse-reward setup and provides an implementation code. Finally, we compared the proposed fusion with the disagreement method using the variance of multiple predicted next states as the intrinsic reward (Pathak et al., 2019).

For fair comparison, we used PPO with the same neural network architecture and common hyperparameters. We also applied the same normalization technique in Appendix C for all the considered intrinsic reward generation methods so that the performance difference results only from the intrinsic reward generation method. In the case of the state-of-the-art algorithm by Zheng et al. (2018), we verified reproducibility for the setup $\Delta = 40$ by obtaining the same result as the reference. (See Appendix D for a detailed description of the overall hyperparameters for simulations and reproducibility.)

Fig. 2 shows the comparison results. It is observed that the proposed fusion-based intrinsic reward generation method yields top-level performance. The gain is significant in Hopper and Walker2d, and the performance variance is much smaller than the state-of-the-art intrinsic reward module method in most cases.

## 4.2 ABLATION STUDY

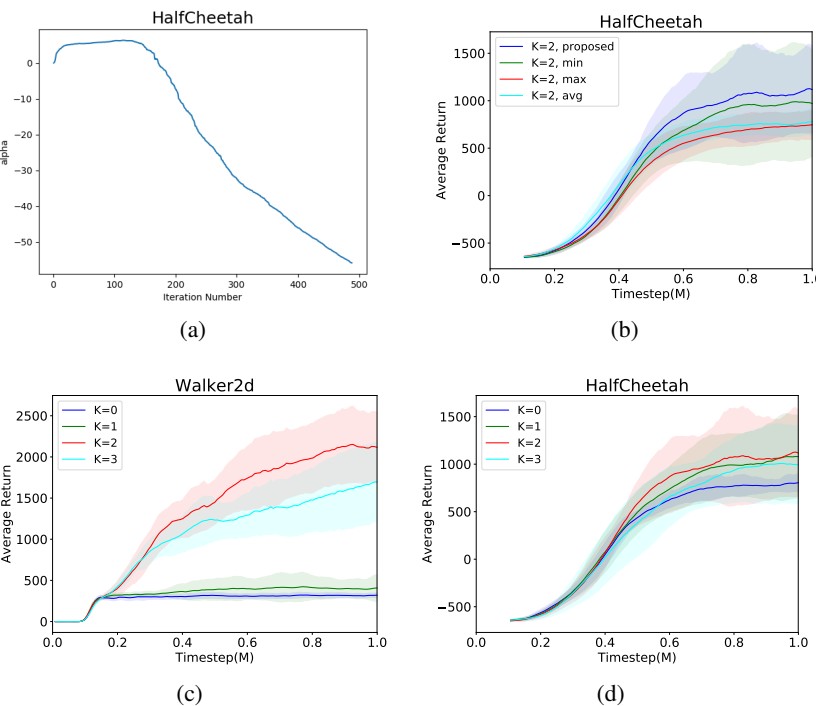

Figure 3: (a) Learning curve of $\alpha$ during the proposed fusion learning in HalfCheetah for 1M timesteps. (b) The performance comparison with static fusion methods. (c, d) Mean performance for 1M timesteps as a function of $K$ for (c) Walker2d and (d) HalfCheetah. $K = 0$ means PPO without intrinsic reward, and $K = 1$ means the single-model surprise method. ($K = 4$ yielded similar performance to that of $K = 3$, so we omitted the curve of $K = 4$ for simplicity.)

### 4.2.1 LEARNING BEHAVIOR OF FUSION PARAMETER $\alpha$

We investigated how the fusion parameter $\alpha$ changed adaptively during the training. Fig. 3(a) shows the learning curve of the fusion parameter $\alpha$ in HalfCheetah. It is seen that starting from the initial value $\alpha = 0$, the fusion parameter $\alpha$ increases until it reaches approximately 5, maintains the level until approximately 180 iterations (0.4 million timesteps), and then decreases monotonically. The proposed fusion learning method takes relatively more aggressive fusion strategies with $\alpha$ being around 5 (but this is not the too aggressive maximum corresponding to $\alpha = \infty$) in the early stage of learning. Then, the fusion learning takes more and more conservative fusion strategy by decreasing $\alpha$ more and more to large negative values (i.e., towards minimum taking). This observation is consistent with the general behavior of RL that aggressive exploration is essential in the early stage of learning and conservative exploitation has a more considerable weight in the later stage of learning.

As seen in Fig. 3(b), in the fixed fusion case, the method using the average has higher performance than that with minimum or maximum in the early stage of training. However, the minimum selection method yields better performance than average or maximum at the later stage. It is seen that the proposed adaptive fusion yields the best performance because the proposed adaptive fusion takes advantage of both fast performance improvement in the early stage and high final performance at the end by learning $\alpha$ optimally.

In order to see the difference between the proposed $\alpha$-fusion learning and other fusion learning method, we considered a fusion method directly using neural networks. In the considered method, we designed a neural network fusion function $f_\xi(x_1, \cdots, x_K)$ of $K$ inputs with (i) linear activation or (ii) nonlinear ($\tanh$) activation. In both cases, $f_\xi$ has a single hidden layer of size $2K$. It is observed that our proposed method outperforms the fusion with learned neural networks using the same KLD model error input. See Appendix E for the comparison result.

### 4.2.2 EFFECT OF THE NUMBER OF PREDICTION MODELS

We investigated the impact of the model order $K$. Since we adopt Gaussian distributions for the prediction models $P_{\phi^1}, \cdots, P_{\phi^K}$, the mixture $P_K$ is a Gaussian mixture for given state-action pair $(s, a)$. According to a recent result (Haarnoja et al., 2018), the model order of a Gaussian mixture need not be too large to capture the true transition probability distribution effectively in practice. Thus, we evaluated the performance for $K = 1, 2, 3, 4$. Fig. 3(c) and 3(d) show the mean performance as a function of $K$ in Walker2d and HalfCheetah. The performance improves as $K$ increases. Once the proper model order is reached, the performance does not improve further due to more difficult model estimation for higher model orders, as expected from our intuition. From this result, we found that $K = 2$ or 3 seems proper for all the six environments considered in Section 4.1.

## 5 CONCLUSION

In this paper, we proposed a new adaptive fusion method with multiple prediction models for sparse-reward RL. The mixture of multiple prediction models is used to better approximate the unknown transition probability, and the intrinsic reward is generated by adaptive fusion learning with multiple prediction error values. The ablation study shows that the general principle of RL is valid even in the adaptive fusion that we need to take a more aggressive strategy in the early stage and less aggressive strategy in the later stage. Numerical results show that the proposed method outperforms existing intrinsic reward generation methods in the considered sparse environments. The proposed adaptive fusion structure is useful not only to the specific problem considered here but also to other problems involving numeric fusion with fusion learning.

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

# A   ALGORITHM

---

**Algorithm 1** Sparse RL with Fusion Learning with Multiple Prediction Models for Intrinsic Reward Generation: PPO Case

---

1: $L$ : batch size for policy training, $L'$ : batch size for model training.
2: $L_{mini}$ : minibatch size for policy training, $L'_{mini}$ : minibatch size for model training.
3: $N$ : epoch size for policy training, $N'$ : epoch size for model training.
4: $MAX$ : the maximum index of batch period $l$, $K$ : the number of prediction models.
5: Initialize the policy $\pi_{\theta_0}$, the $K$ transition probability models $P_{\phi_0^1}, \cdots, P_{\phi_0^K}$, the mixing coefficients $q_1, \cdots, q_K$, the fusion parameter $\alpha$, and the extrinsic value function parameter $\zeta$.
6: Generate trajectories with $\pi_{\theta_0}$ and store them to the initially empty replay buffer $M$.
7: **for** Batch period $l = 0, \cdots, MAX - 1$ **do**
8:     Train $P_{\phi_l^1}, \cdots, P_{\phi_l^K}$ by performing gradient updates for (10), and update $q_1, \cdots, q_K$ by performing iterations with (11). For each $P_{\phi_l^i}(1 \leq i \leq K)$, we draw a batch $D'_i$ of size $L'$ randomly and uniformly from $M$, and perform the updates with minibatches of size $L'_{mini}$ drawn from $D'_i$ for $N'$ epochs.
9:     **for** Timestep $t = lL, lL + 1, \cdots, lL + L - 1$ **do**
10:         Collect $s_t$ from the environment and $a_t$ with the policy $\pi_{\theta_l}$.
11:         Collect $s_{t+1}$ and the extrinsic reward $r_t$ from the environment and add $(s_t, a_t, s_{t+1})$ to $M$.
12:     **end for**
13:     Acquire the intrinsic reward $r_{t,int}$ of the current batch $D$ of size $L$. (Detail is described in Appendix C.)
14:     Train $\pi_{\theta_l}$ by using PPO with the total rewards (12) and minibatch size $L_{mini}$ for $N$ epochs.
15:     Train $\alpha$ and $\zeta$ by using the parameter learning method described in Section 3.3 with the total rewards (12) and minibatch size $L_{mini}$ for $N$ epochs.
16: **end for**

---

## B    AN EXAMPLE OF $\alpha$-MEAN WITH VARYING $\alpha$

.

The $\alpha$-mean includes all numeric fusions with the scale-free property such as minimum, maximum, and conventional mean functions such as arithmetic, geometric, and harmonic mean, by varying $\alpha$. For example, the $\alpha$-mean $f_\alpha$ of two values $x_1$ and $x_2$ as a function of $\alpha$ is given as

$$f_\alpha(x_1, x_2) = \begin{cases} \max\{x_1, x_2\} \text{ (maximum)} & \text{if } \alpha = -\infty \\ \frac{x_1+x_2}{2} \text{ (arithmetic)} & \text{if } \alpha = -1 \\ \sqrt{x_1 x_2} \text{ (geometric)} & \text{if } \alpha = 1 \\ \left(\frac{x_1^{-1}+x_2^{-1}}{2}\right)^{-1} \text{ (harmonic)} & \text{if } \alpha = 3 \\ \min\{x_1, x_2\} \text{ (minimum)} & \text{if } \alpha = \infty \end{cases} \quad . \tag{13}$$

The figure below shows the $\alpha$-mean of $x_1 = 1$ and $x_2 = 5$ with respect to $\alpha$. The $\alpha$-mean can implement all the possible fusion of $K$ values into a single value under the condition of scale-freeness by varying $\alpha$.

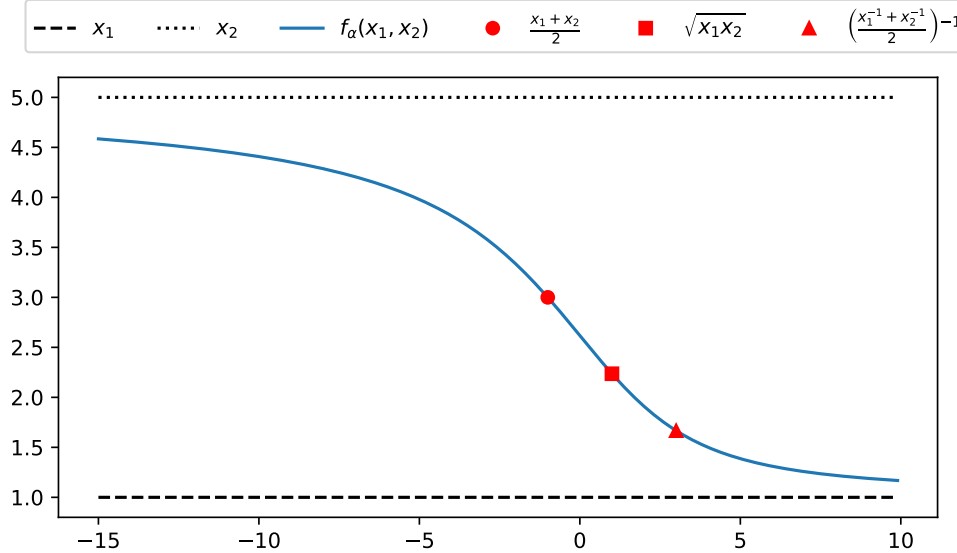

Figure 4: $\alpha$-mean of $x_1 = 1$ and $x_2 = 5$

## C  1-STEP TECHNIQUE AND NORMALIZATION FOR $r_{t,int}^j$

For actual computation of the intrinsic reward, we first applied the 1-step technique (Achiam & Sastry, 2017). Note that

$$r_{t,int}^j(s_t, a_t, s_{t+1}) = \log \frac{P_K(s_{t+1}|s_t, a_t)}{P_{\phi^j}(s_{t+1}|s_t, a_t)} \tag{14}$$

where

$$P_K = \sum_{i=1}^K q_i P_{\phi^i}.$$

With the 1-step technique, the batch index for the numerator in (14) is the current batch index $l(t)$ corresponding to time step $t$, whereas the batch index for the denominator in (14) is the previous batch index $l(t) - 1$. Hence, with the 1-step technique, $r_{t,int}^j(s_t, a_t, s_{t+1})$ is modified as

$$\log \frac{\sum_{i=1}^K q_i P_{\phi_{l(t)}^i}(s_{t+1}|s_t, a_t)}{P_{\phi_{l(t)-1}^j}(s_{t+1}|s_t, a_t)}. \tag{15}$$

Then, this modified $j$-th model's prediction error value is applied to the $\alpha$-fusion in Section 3.3.

To improve numerical stability, we further applied the normalization technique (Achiam & Sastry, 2017) to the output $f_\alpha(s_t, a_t, s_{t+1})$ of the $\alpha$-fusion. Thus, the final actual intrinsic reward given the current batch $D$ is expressed as

$$r_{t,int}(s_t, a_t, s_{t+1}) = \frac{f_\alpha(s_t, a_t, s_{t+1})}{\max\left\{ \frac{|\sum_{(s,a,s') \in D} f_\alpha(s,a,s')|}{|D|}, 1 \right\}}, \tag{16}$$

where (16) describes the applied normalization.

## D  NEURAL NETWORK ARCHITECTURE AND HYPERPARAMETERS

For actual implementation, the code implemented by Dhariwal et al. (2017) and Zheng et al. (2018) are used. The policy, the prediction models, the value function of total reward, and the value function of extrinsic reward $V_\zeta$ were designed by fully-connected neural networks, all of which had two hidden layers of size (64, 64) (Houthooft et al., 2016; Dhariwal et al., 2017; Tang et al., 2017). The tanh activation function was used for all of the networks (Achiam & Sastry, 2017; Dhariwal et al., 2017). The means of the fully factorized Gaussian prediction models were the outputs of our networks, and the variances were trainable variables that were initialized to 1 (Dhariwal et al., 2017). Other than the variances, all initialization is randomized so that each of the prediction models was set differently (Kurutach et al., 2018; Tavakoli et al., 2018). For the implementation of the policy model and the value function, our method and all the considered intrinsic reward generation method used the same code for the module method (Zheng et al., 2018).

Although a recent work (Achiam & Sastry, 2017) used TRPO (Schulman et al., 2015a) as the baseline learning engine, we used PPO (Schulman et al., 2017), one of the currently most popular algorithms for continuous action control, as our baseline algorithm. While the same basic hyperparameters as those in the previous work (Achiam & Sastry, 2017) were used, some hyperparameters were tuned for PPO. $\lambda$ for the GAE method (Schulman et al., 2015b) was fixed to 0.95, while the discounting factor was set to $\gamma = 0.99$. The batch size $L$ for the training of the policy was fixed to 2048. For the policy update using PPO, the minibatch size $L_{mini}$ was set to 64, the epoch number $N$ 10, the value function coefficient 0.5, the clipping constant 0.2, and the entropy coefficient 0.0. The initial values of the learning rates of Adam optimizer (Kingma & Ba, 2014) for updating the policy parameter $\theta$ and the extrinsic value function parameter $\zeta$ were fixed to 0.0003 and 0.0001, respectively. The initial value of the learning rate of $\alpha$ was 0.01 for Hopper, HalfCheetah, Humanoid, and InvertedPendulum, and 0.001 for Ant and Walker2d. The three learning rates were linearly decayed as timestep passed so that the values at the end of the training was 0. The initial value of $\alpha$ was set to 0.

Each of the single-model surprise method, the hashing method, and our proposed method requires a replay buffer. The size of the used replay buffer for all these three methods is 1.1M. If the buffer becomes full, the earlier samples are deleted first. Before the beginning of the iterations, $2048 \times B$ samples from real trajectories generated by the initial policy were added to the replay buffer. We set $B = 40$ for our experiments. For the methods not requiring a replay buffer, i.e., Curiosity, Information Gain, Module, Disagreement, and PPO Only, we ran $2048 \times B = 81920$ timesteps before measuring performance for fair comparison.

For the prediction model learning, we set the batch size $L' = 2048$, $L'_{mini} = 64$, and $N' = 4$. The optimization (10) was solved based on second-order approximation (Schulman et al., 2015a). When $K = 1$, the optimization (10) reduces to the model learning problem in Achiam & Sastry (2017). In Achiam & Sastry (2017), the constraint constant $\kappa$ in the second-order optimization was well-tuned as 0.001. Therefore, we used this value of $\kappa$ not only to the case of $K = 1$ but also to the case of $K \geq 2$. We further tuned the value of $c_{reg}$ in (10) for each environment, and we set $c_{reg} = 0.01$ . For the information gain method, we need another hyperparameter $h$ which is the weight to balance the original intrinsic reward and the homeostatic regulation term (de Abril & Kanai, 2018). We tuned this hyperparameter for each environment and the used value of $h$ is shown in Table 1. For the disagreement method, we used five deterministic models.

| | Ant | Hopper | HalfCheetah | Humanoid | InvertedPendulum | Walker2d |
|---|---|---|---|---|---|---|
| Curiosity | $\beta = 0.01$ | $\beta = 0.01$ | $\beta = 0.0001$ | $\beta = 0.1$ | $\beta = 0.003$ | $\beta = 0.03$ |
| Hashing | $\beta = 0.0001$ | $\beta = 0.01$ | $\beta = 0.00001$ | $\beta = 0.1$ | $\beta = 0.0001$ | $\beta = 0.003$ |
| Information Gain | $\beta = 0.01, h = 4$ | $\beta = 0.1, h = 4$ | $\beta = 0.0001, h = 4$ | $\beta = 0.01, h = 2$ | $\beta = 0.0001, h = 4$ | $\beta = 0.03, h = 2$ |
| Single Surprise | $\beta = 0.00001$ | $\beta = 0.02$ | $\beta = 0.0003$ | $\beta = 0.1$ | $\beta = 0.0001$ | $\beta = 0.02$ |
| Disagreement | $\beta = 0.00003$ | $\beta = 0.003$ | $\beta = 0.003$ | $\beta = 0.0003$ | $\beta = 0.001$ | $\beta = 0.1$ |

Table 1: Used hyperparameter values.

Table 1 summarizes the weighting factor $\beta$ as well as the hyperparameter $h$ in information gain method. The weighting factor $\beta$ in (12) between the extrinsic reward and the intrinsic reward should be determined for all intrinsic reward generation methods. Since each of the considered methods yields different scale of the intrinsic reward, we used the optimized weighting factor $\beta$ in (12) for each algorithm for each environment by testing $\beta$ according to log scale (Zheng et al., 2018):

$\{1.0, 0.5, 0.3, 0.2, 0.1, \cdots, 10^{-6}\}$. In the single-model surprise method and the proposed method, the proposed method employed the same hyperparameters as the single-model surprise method. We confirmed that the hyperparameters of the other five methods were well-tuned in the original papers (Pathak et al., 2017; Tang et al., 2017; de Abril & Kanai, 2018; Zheng et al., 2018; Pathak et al., 2019), and we used the hyperparameters provided by these methods.

For the intrinsic reward module method, we checked that the open-source code reproduced results in Zheng et al. (2018), as shown in Fig. 5. 'Module 0.01' represents the module method with training using the sum of intrinsic reward and the scaled extrinsic reward with scaling factor 0.01. 'Module 0' represents training using intrinsic reward only (no addition of extrinsic reward). Both methods are introduced in Zheng et al. (2018), and we checked reproducibility when $B = 0$, i.e., we ran $2048 \times B = 0$ timesteps before measuring performance. We observed that our used code yielded the same results as those in Zheng et al. (2018) (InvertedPendulum is not considered in this paper).

Thus, we used this code for the module method with only one change that we ran $2048 \times B$ timesteps with $B = 40$ before measuring performance for fair comparison. (Since the range of intrinsic reward from the module method is $[-1, 1]$, intrinsic reward normalization in (16) is not needed.) For the module method in performance comparison 4.1, we selected a better method between 'Module 0' and 'Module 0.01', assuming $B = 40$. 'Module 0' performed better than 'Module 0.01' in Hopper and Walker2d, and 'Module 0.01' performed better than 'Module 0' in the other three environments. We also observed that both 'Module 0' and 'Module 0.01' performed poorly in InvertedPendulum, which is not considered in Zheng et al. (2018). Therefore, we further fine-tuned both of the two scaling coefficients of extrinsic and intrinsic reward and set the best values of $1.0$ and $0.003$, respectively.

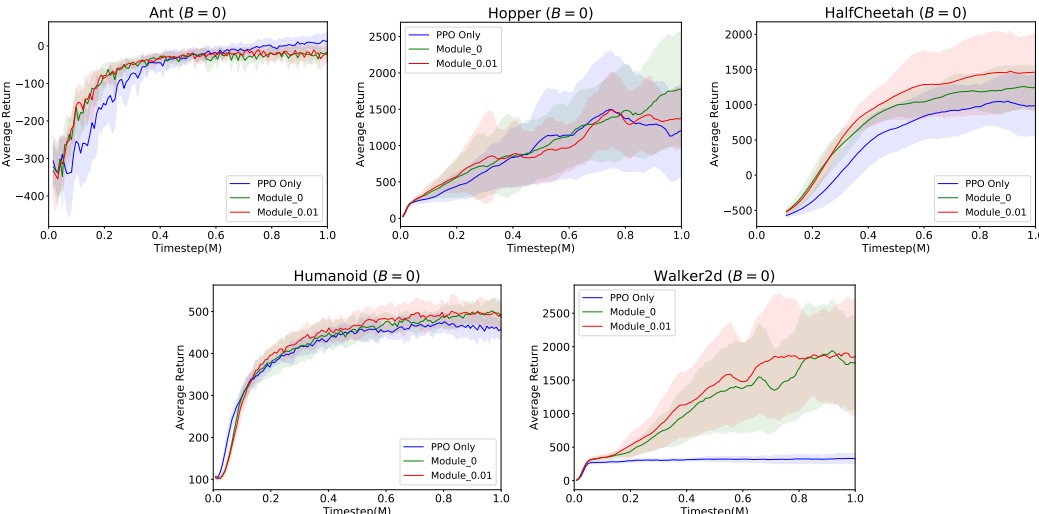

Figure 5: Reproduced mean performance of the module method over 10 random seeds with $\Delta = 40$ when $B = 0$.

# E    COMPARISON TO DIRECT NEURAL NETWORK-BASED FUSION

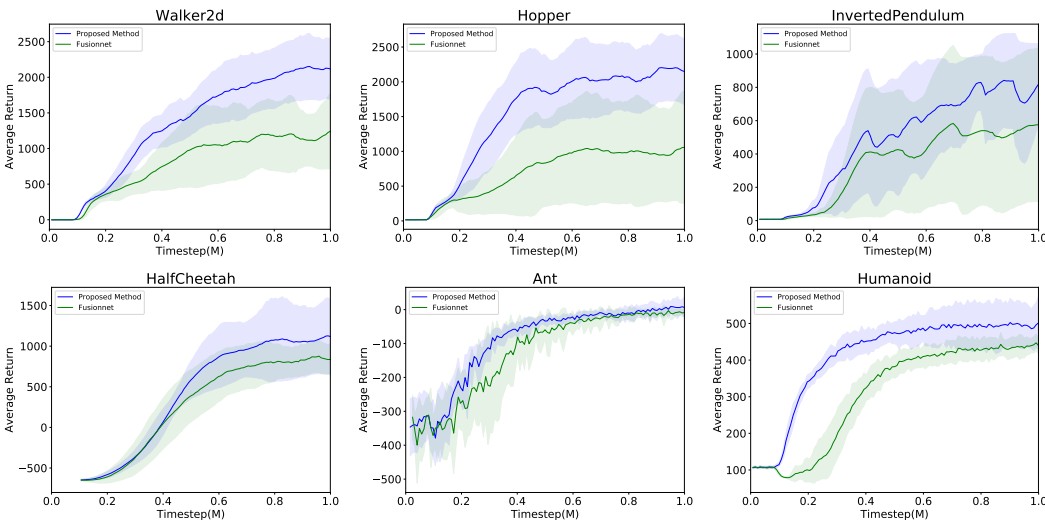

Figure 6:   Performance comparison between our proposed method (blue curve) and fusion with neural network learning with nonlinear activation (green) when $K = 2$. (Fusion with neural network learning with linear activation performed worse than the nonlinear activation, so we omitted the result.) All simulations were conducted over ten fixed random seeds.

In order to compare our fusion method to the fusion with neural network learning, we designed a neural network fusion function $f_\xi(x_1, \cdots, x_K)$ of $K = 2$ inputs with (i) linear activation or (ii) nonlinear ($\tanh$) activation. In both cases, $f_\xi$ has a single hidden layer of size $2K$. Note that our fusion function with a learnable $\alpha$ is explicitly expressed as $f(x, y) = -\frac{2}{1-\alpha} \log\left[\frac{\exp\left(-\frac{1-\alpha}{2}x\right) + \exp\left(-\frac{1-\alpha}{2}y\right)}{2}\right]$ for KLD error approximation inputs by the exponentiation at the input stage. Fig. 6 shows that our method outperforms the fusion with neural network learning using the same KLD model error input.

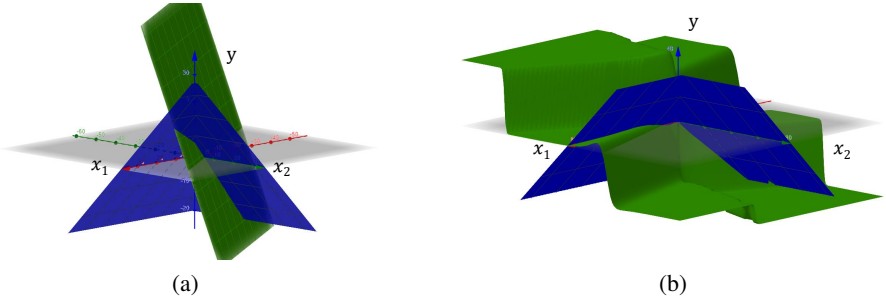

Figure 7:   Visualizations of our fusion function and the fusion method based on neural network learning (both were trained in the Hopper environment). The gray surface is the $(x_1, x_2)$ input plane, and and the $y$-axis is the fusion output: (a) the proposed fusion function (blue) and the linear neural network fusion function (green). (b) The proposed fusion function (blue) and the nonlinear neural network fusion function (green).

In Fig. 7, we plot both the proposed fusion function $y = f_\alpha(x_1, x_2)$ and the neural-network-based fusion function $y = f_\xi(x_1, x_2)$ after training in the Hopper environment. Figs. 7(a) and 7(b) shows the comparison in the linear and non-linear hyperbolic tangent activation cases, respectively. As expected, in the linear activation case, $(x_1, x_2, y)$ forms a hyperplane. Although the hyperplane is fit in a best way, still it cannot perform properly as a fusion function. For example, permutation invariance is a property of a good fusion function. However, a single linear layer network cannot achieve this property although the proposed $\alpha$-fusion has the permutation invariance property. Now,

consider the case of non-linear hyperbolic tangent activation. In this case, as seen in Fig. 7(b), the learned fusion function seems a bit better than the linear activation case but is not still symmetric around the line $x_1 = x_2$. Hence, it is not permutation invariant either. It seems that the neural-network-based fusion function requires more complexity and more learning time. The proposed adaptive $\alpha$-fusion structure captures the fusion behavior only by using a single parameter $\alpha$ and the corresponding learning is efficient.

# F    FUSION WITH DIFFERENT PREDICTION ERRORS: THE MEAN SQUARED ERROR (MSE) CASE

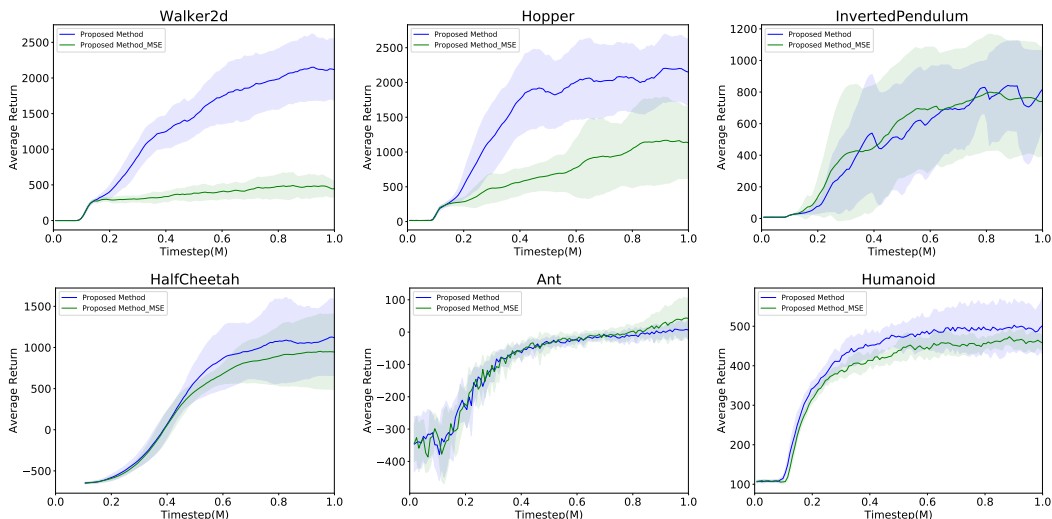

Figure 8: Performance comparison between the proposed method using KLD (blue) and the proposed method using MSE (green). All simulations were conducted over ten fixed random seeds.

We compared the proposed method using KLD with the proposed method using MSE. In the MSE method, we trained ten deterministic models predicting the next state. These multiple models are differently initialized and independently trained with different batch data as widely done in other works (Nagabandi et al., 2017; Kurutach et al., 2018). Then, we calculated the MSE between the actual next state and the predicted next state as the error measure.

Fig. 8 shows that the proposed method using KLD outperforms the proposed method using MSE. Since the KLD uses joint training among multiple probabilistic models, the KLD method can reflect the underlying dynamics more accurately. On the other hand, independently trained deterministic models were not diverse enough to effectively capture the underlying dynamics as compared to the joint training.

## G   NECESSITY OF SCALE-FREE PROPERTY (CONDITION 2)

Note that basically the fusion function is some kind of averaging function. Condition 2 is a proper condition for any reasonable averaging function (Hardy et al., 1952). The scale-free property is also called the homogeneous property.

Suppose that the homogeneous property for the function function $f$ is not satisfied and assume the nonhomogeneous relationship:

$$f(cx_1, cx_2) = c'f(x_1, x_2), \tag{17}$$

where $c'$ $(\neq c)$ is mapped for each scaling factor $c$ for the completeness of the scaling operation.

Now suppose that there exist two constants $c, c'$ such that $c > 1$ and $0 < c' < 1$ and $f(cx_1, cx_2) = c'f(x_1, x_2)$ for two input $x_1$ and $x_2$. Then, the monotonicity is broken. The two inputs $x_1$ and $x_2$ are increased as $cx_1$ and $cx_2$ but the corresponding output is reduced as $c'f(x_1, x_2)$ as compared to the original output $f(x_1, x_2)$. This is not the situation that we want. Furthermore, by repeatedly applying equation 17, we have

$$\lim_{n \to \infty} [f(c^n x_1, c^n x_2) = (c')^n f(x_1, x_2)] \tag{18}$$

yielding

$$f(\infty, \infty) = 0 \cdot f(x_1, x_2) = 0. \tag{19}$$

So, we have a contradiction. In the case of $0 < c < 1$ and $c' > 1$, we have a similar contradiction:

$$f(0, 0) = \infty \cdot f(x_1, x_2) = \infty. \tag{20}$$

Now assume that there exist $c, c' > 1$ and $c \neq c'$. Set two inputs as $x_1 = x_2 = x$. Then, we have

$$g(x) \stackrel{\triangle}{=} f(x, x) \tag{21}$$

$$g(cx) = f(cx, cx) = c'f(x, x) = c'g(x) \tag{22}$$

By repeating the iteration, we have

$$g(c^n x) = (c')^n g(x). \tag{23}$$

In equation (23), the input to the function $g(x)$ is exponentially increasing as $c^n$ and the output increases exponentially as $(c')^n$. Such function $g(x)$ is uniquely given by the form

$$g(x) \sim (c')^{\log_c x}, \tag{24}$$

where $\sim$ means the scaling equivalence. However, for a different pair $\tilde{c}$ and $\tilde{c}'$, we also require

$$g(x) \sim (\tilde{c}')^{\log_{\tilde{c}} x}. \tag{25}$$

The two functions in (24) and (25) cannot be the same in general. Furthermore, $g(x)$ should be the same for all pairs $(c, c')$. This cannot be satisfied in general. So, we have an indiscrepancy in the nonhomogeneous case. A similar situation happens for $0 < c, c' < 1$. However, note that if we have $c = c'$ for all scaling factor $c$, then the two functions in (24) and (25) are consistent. In this case, we have

$$g(x) \sim (c)^{\log_c x} = x, \tag{26}$$

and this makes sense because if the input values are all the same, the output should be the same as the input. Please note that the generalized mean or $\alpha$-mean exactly satisfies the scaling behavior (26).

## H  EXPLORATION WITHOUT INTRINSIC REWARD GENERATION

Recent indirect methods can further be classified mainly into two groups: (i) revising the original objective function to stimulate exploration, exploiting intrinsic motivation implicitly, and (ii) perturbing the parameter space of policy.

In the first group, Andrychowicz et al. (2017) suggested additional sampling states from the replay buffer and setting those data as new goals for sparse and binary extrinsic reward environments. In their work, the policy was based on the input of both state and goal. In our work, on the other hand, the goal concept is not necessary. Oh et al. (2018) proposed that exploration can be stimulated by exploiting novel state-action pairs from the past and used sparse-reward environments by delaying extrinsic rewards. Hong et al. (2018) revised the original objective function for training by considering the maximization of the divergence between the current policy and recent policies, with an adaptive scaling technique. The dropout concept was applied to the PPO algorithm to encourage the stochastic behavior of the agent episode-wisely (Xie et al., 2019). Convex combination of the target policy and any given policy is considered a new exploratory policy (Shani et al., 2019), which corresponds to solving a surrogate Markov Decision Process, generalizing usual exploration methods such as $\epsilon$-greedy or Gaussian noise.

In the second group, Colas et al. (2018) proposed a goal-based exploration method for continuous control, which alternates generating parameter-outcome pair and perturbing specific parameters based on randomly drawn goal from the outcome space. Recently, inspired by Chua et al. (2018), Shyam et al. (2019) considered pure exploration MDP without any extrinsic reward with the notion of utility, where utility is based on JSD and the Jensen-Rényi divergence (Rényi et al., 1961). They considered several models for the transition functions in this work, but they used this to compute utility based on the multiple models' average entropy.

