# OpenReview forum: "Adaptive Multi-model Fusion Learning for Sparse-Reward Reinforcement Learning"
_ICLR.cc/2021/Conference — Reject_

### Official Review · AnonReviewer1 · 2020-10-28
**A good paper with a missing baseline**

**Rating:** 7
**Confidence:** 4

**Review:**

This paper proposes an intrinsic reward formulation to address the challenge of sparse reward in reinforcement learning. The key idea is to learn multiple models. The prediction error of each model is used as a component of the intrinsic reward. These components are fused using an alpha-mean function. The parameter alpha can be tuned automatically throughout the training process using meta-gradient methods. The method is evaluated on 6 OpenAI continuous control benchmarks (with delayed reward), and demonstrates better performance than several state-of-the-art prior works on intrinsic reward.

Overall, the paper is tackling an important challenge of reinforcement learning. Thus its potential impact is large. The paper is also well written and the results are promising. My main concern is that one important baseline is missing: [Pathak 2019]: "Self-supervised exploration via disagreement." This prior work is very relevant. It also used multiple models and the implementation was so much simpler than the proposed method in this paper. In addition, this paper also claims to be a generalization of [Pathak 2019]. Thus, adding [Pathak 2019] into the baselines for comparisons seems necessary.

I am also curious why not use MSE to measure the model-prediction error? The KL Divergence used in this paper introduces a lot of complexities, both in derivation and implementation (e.g. eq. 9, 10, 11). If MSE is not compatible with the proposed method, more discussions in the paper is welcome. Otherwise, adding an ablation to compare MSE with KLD would help, especially if KLD yields better results than MSE. This would justify the extra complexity introduced by using KLD.

The need of Condition 2 in Section 3.3 is not sufficiently motivated. How does this condition affect derivation, implementation and final results? It would be great to have a few sentences to clarity this.

In summary, this is a good paper. However, missing an important baseline [Pathak 2019] seems a significant oversight. I hope that this baseline will be added in the future version of this paper.

---

> ### Author Response · Authors · 2020-11-24
> **Response to Reviewer 1**
>
> We thank you for your valuable comments and constructive feedback.
>
> ### 1. Additional baseline
>
> On page 6 in the main paper, the performance plots of additional baseline [Pathak et al., 2019] (suggested by the reviewer) are added in Figure 2 of the new version as yellow lines denoted as ‘Disagreement.’ It is observed that the proposed adaptive fusion method outperforms this baseline over the considered six environments.
>
> [Ref Pathak et al., 2019] Pathak et al., "Self-supervised exploration via disagreement," ICML 2019.
>
>
> ### 2. The proposed method with mean-squared error (MSE)
>
> We compared the proposed method using KLD with the proposed method using MSE. In the MSE method, we trained ten deterministic models predicting the next state. These multiple models are differently initialized and independently trained with different batch data as widely done in other works [Nagabandi et al., 2017], [Kurutach et al., 2018]. Then, we calculated the MSE between the actual next state and the predicted next state as the error measure. It is observed that the proposed fusion with KLD outperforms the proposed fusion method using MSE (Fig. 8 in Appendix F, on page 19 of the new version).
>
> Since the KLD uses joint training among multiple probabilistic models, the KLD method can reflect the underlying dynamics more accurately. On the other hand, independently trained deterministic models were not diverse enough to effectively capture the underlying dynamics compared to the joint training.
>
>
> [Ref Nagabandi et al., 2017] Nagabandi et al., "Neural network dynamics for model-based deep reinforcement learning with model-free fine-tuning," arXiv 2017.
>
> [Ref Kurutach et al., 2018] Kurutach et al., "Model-ensemble trust-region policy optimization," ICLR 2018.
>
>
>
>
> ### 3. The necessity of scale-free property (Condition 2)
>
> Note that basically, the fusion function is some averaging function. Condition 2 is a proper condition for any reasonable averaging function [Hardy et al., 1952]. The scale-free property is also called the homogeneous property.
>
> Suppose that the homogeneous property for the function function $f$ is not satisfied and assume the nonhomogeneous relationship $f(c x_1, c x_2) = c^\prime f(x_1, x_2)$,
> where  $c^\prime ~(\ne c)$ is mapped for each scaling factor $c$ for the completeness of the scaling operation.
>
>
> Now suppose that there exist two constants $c,c^\prime$ such that $c> 1$ and $0<c^\prime <1$ and  $ f(c x_1, c x_2) = c^\prime f(x_1, x_2)$
> for two input $x_1$ and $x_2$. Then, the monotonicity is broken.  The two inputs $x_1$ and $x_2$ are increased as $cx_1$ and $c x_2$ but the corresponding output is reduced as $c^\prime f(x_1, x_2)$ as compared to the original output $f(x_1,x_2)$. This is not the situation that we want. Furthermore, by repeatedly applying $ f(c x_1, c x_2) = c^\prime f(x_1, x_2)$, we have
> \begin{equation}
> \lim_{n\rightarrow \infty} [f(c^nx_1,c^nx_2) = (c^\prime)^n f(x_1,x_2)]
> \end{equation}
> yielding
> \begin{equation}
>     f(\infty,\infty) = 0\cdot f(x_1,x_2) = 0.
> \end{equation}
> So, we have a contradiction.  In the case of $0 < c< 1$ and $c^\prime >1$, we have a similar contradiction:
> \begin{equation}
>     f(0,0) = \infty \cdot f(x_1,x_2) = \infty.
> \end{equation}
>
> Now assume that there exist $c, c^\prime >1$ and $c \ne c^\prime$. Set two inputs as $x_1=x_2=x$. Then, we have
> \begin{eqnarray}
> g(x) &\stackrel{\triangle}{=}& f(x,x), \\
> g(cx) &=& f(cx,cx) = c^\prime f(x,x)= c^\prime g(x).
> \end{eqnarray}
> By repeating the iteration, we have
> \begin{equation}
>     g(c^n x) = (c^\prime)^n g(x).
> \end{equation}
>
>
>
> In this equation, the input to the function $g(x)$ is exponentially increasing as $c^n$ and the output increases exponentially as $(c^\prime)^n$. Such function $g(x)$ is uniquely given by the form
> \begin{equation}
> g(x) \sim (c^\prime)^{\log_c x},
> \end{equation}
> where $\sim$ means the scaling equivalence.
> However, for a different pair $\tilde{c}$ and $\tilde{c}^\prime$, we also require
> \begin{equation}
> g(x) \sim (\tilde{c}^\prime)^{\log_{\tilde{c}} x}.
> \end{equation}
> The two functions above  cannot be the same in general.
> Furthermore, $g(x)$ should be the same for all pairs $(c,c^\prime)$. This cannot be satisfied in general.
> So, we have an indiscrepancy in the nonhomogeneous case. A similar situation happens for $c,c^\prime < 1$. However, note that if we have $c=c^\prime$ for all scaling factor $c$, then the two functions $(c^\prime)^{\log_c x}$ and $(\tilde{c}^\prime)^{\log_{\tilde{c}} x}$ are consistent. In this case, we have
> \begin{equation}
>     g(x) \sim (c)^{\log_c x} =x,
> \end{equation}
> and this makes sense because if the input values are all the same, the output should be the same as the input. Please note that the generalized mean or $\alpha$-mean exactly satisfies the scaling behavior.
>
>
> [Ref. Hardy et al., 1952] Hardy et al., "Inequalities," Cambridge Mathematical Library, 1952.

---

> > ### Comment · AnonReviewer1 · 2020-11-25
> > **Thanks for your response**
> >
> > Your response and the new experiments answer my questions. I will update my score accordingly.

---

### Official Review · AnonReviewer4 · 2020-10-28
**This paper proposes a new method to fuse predictions from distinct models in the sparse-reward reinforcement learning scenario.**

**Rating:** 5
**Confidence:** 3

**Review:**

In this paper, the authors present a generalization of the model-prediction-error-based intrinsic reward method by fusing predictions from multiple models.
The authors considered the sparse reward scenario in reinforcement learning.

In related work, the authors mentioned that previous works on image spaces are not directly related to theirs. However, I did not understand what are the limitations or the caveats of the proposed method that leads to this conclusion.

Also in related work, I did not understand the purpose of detailing not related work in Sec. 2.2., I believe that the authors could use this space to discuss, for instance,
applications of fusion methods in reinforcement learning scenarios, such as: “Data fusion using Bayesian theory and reinforcement learning method”.
Another option, even better, is discussing approaches proposed in the line of investigation using ensembles
(e.g. “Model Ensemble-Based Intrinsic Reward for Sparse Reward Reinforcement Learning”).

I would like to understand the computational costs involved in using such a fusion approach., both in terms of individual methods and alpha optimization.

In Fig. 2, it is hard to conclude that the gain over Module baseline is significant only considering the figure. I don’t know if the authors are considering some
statistical test when they mention significance, if this is the case, they should properly present the test and premisses.

In Sec.4.2.4, it is hard to see how the performance improves as K increases. First, only four values for K are a limitation to conclude this.
Additionally, for instance, when considering the Walker2d dataset, the performance for K=2 is better than K=3.

References should point to the published work rather than the arxiv entries, when the former is available (e..g. Exploration by random network distillation, ICLR’19).

---

> ### Author Response · Authors · 2020-11-24
> **Response to Reviewer 4**
>
> Thank you for your various comments on helping to improve the quality of our paper.
>
>
> ### 1. Related work (section 2): Why did we deal with indirect exploration (section 2.2)?
>
> Because indirect exploration is another main branch in intrinsically motivated RL, we included this part to complement the related work. As suggested by the reviewer, we will consider omitting this part and include more results obtained during the review period.  Also, we updated references to the published work in the last version.
>
>
>
> ### 2. Model order and computational costs
>
> We designed our model as a Gaussian mixture for a given state-action pair. A recent paper [Haarnoja et al., 2018] explains that a Gaussian mixture with a practical model order such as two or four can sufficiently represent the underlying distribution for a medium-dimensional output space. We verified that the model order of four was sufficient for performance improvement compared to the single-model method.
> The performance does not improve further  (e.g., Walker2d) as we increase the model order further beyond four.  Please note that the computational cost increases only linearly for the model order. Furthermore, a really high model order is not required, as mentioned in  [Haarnoja et al., 2018]. So, multi-model fusion can be doable.
>
>
> [Ref Haarnoja et al., 2018] Haarnoja et al., "Soft actor-critic: Off-policy maximum entropy deep reinforcement learning with a stochastic actor," ICML 2018.
>
>
>
> ### 3. Statistical test on the significance
>
> To mathematically check our method’s gain over the module baseline, we applied the Welch’s t-test on our data at the end of the training. Welch’s t-test is adequate in our setting since we conduct randomized experiments (i.e., every experiment with a specific seed is independent). This test does not require the variance of the two tested methods to be the same.  Furthermore, Welch’s t-test is robust compared to other tests to compare two different RL algorithms [Colas et al., 2019].
>
> We calculated the mean of the proposed method, the mean of the module method, the t-value, and the right-sided p-value. The values are shown below.
> The order of values for each environment is the mean of the proposed method, the mean of the module method, the t-value, and the right-sided p-value.
>
>
>
> Ant: 7.53, -21.74, t-value=2.8941, p-value=$\mathbf{0.0048}$
>
> Hopper: 2065.0, 1427.5, t-value=2.4957, p-value=$\mathbf{0.0116}$
>
> Walker2d: 1646.0, 1043.3, t-value=2.5288, p-value=$\mathbf{0.0126}$
>
> InvertedPendulum: 803.8, 762.1, t-value=0.2936, p-value=0.3862
>
> HalfCheetah: 1120.0, 1158.9, t-value=-0.2288, p-value=0.5882
>
> Humanoid: 495.8, 505.4, t-value=-0.4111, p-value=0.6565
>
> According to Welch's t-test, if the p-value is low near zero, then the confidence in the difference in the two mean values is very high. On the other hand, if the p-value is around 0.5, then the confidence in the two mean values' difference is low.
> As seen in the above values, with high confidence, the proposed method outperforms the module baseline in Ant, Hopper, and Walker2d.  With low confidence, the proposed method outperforms the module baseline in InvertedPendulum.  In HalfCheetah and Humanoid, the proposed method slightly performs worse than the module baseline, but the confidence for this result is very low, as seen in their p-values of 0.5882 and 0.6565.
>
>
>
> [Ref Colas et al., 2019] Colas et al., “A Hitchhiker’s Guide to Statistical Comparisons of Reinforcement Learning Algorithms,” accepted to ICLR 2019 Workshop RML, 2019.

---

### Official Review · AnonReviewer2 · 2020-10-29
**The paper proposed a novel method that fuses multiple transition models to generate intrinsic rewards for sparse reward reinforcement learning**

**Rating:** 6
**Confidence:** 3

**Review:**

In this paper, the authors explore a model based intrinsic reward generation mechanism, in environment settings where the reward assignment is sparse. The authors used an ensemble of models,  and computed the alpha-mean value of their KL divergences with respect to the "true transitions". The alpha-mean serves as the intrinsic reward, with the parameter alpha being co-optimized during the training.  The authors demonstrated that their proposed approach yields top performance in six augmented MuJuCo continuous control tasks.


Pros:

(1) A clearly written paper and easy to read and understand.

(2) The approach demonstrated consistent top performance in all benchmarks.

Cons:

(1) Although the author provides a few ablation studies. I propose to add at least one more study that compares the effectiveness of their method under different "sparsity" settings. In all experiments, the authors used a fixed interval of 40, at which extrinsic rewards are computed. I am interested to see if their conclusion can still hold when this interval is set at, for example, 10, 20, 60, etc.

(2) The alpha-mean method is one type of scale invariant transformation. I wonder if the authors have studied other transformation functions, or even scale invariant kernel functions?

(3) There are other sparse-reward tasks such as pushing/sliding/pick-and-place, which are under a different category than MuJuCo continuous control tasks. I am interested in seeing if the proposed approach can still perform in these settings. This will also help make the author's statements more convincing.

Conclusion:
Please address my concerns raised in the "cons" section.

---

> ### Author Response · Authors · 2020-11-24
> **Response to Reviewer 2**
>
> We thank you for your valuable feedback and constructive comments.
>
>
>
> ### 1. Regarding environments
>
> We considered another sparse reward environment AntMazeSparse [Zhang et al., 2020]. In AntMazeSparse, an Ant robot in a U-shaped maze tries to achieve the target position called 'goal,' and the ant is rewarded by one only if it reaches the goal. We applied our method to AntMazeSparse and observed that our method did not perform well in  AntMazeSparse.  We will consider the extension of our method to the goal-conditioned RL setting as our future work. However, we performed additional experiments on delta=20 and 60, which is the ‘easier’ and ‘harder’ delayed reward task. We observed that our significant improvement in Hopper and Walker2d still holds in both cases compared to the module method.
>
>
> [Ref Zhang et al., 2020] Zhang et al., "Generating Adjacency-Constrained Subgoals in Hierarchical Reinforcement Learning," accepted to NeurIPS 2020.
>
>
>
>
>
> ### 2. Is there any other scale-invariant transformation? Is there any other useful transformation?
>
> [Hardy et al. 1952],[Amari 2007, 2016] proved the generalized mean, i.e., $\alpha$-mean is the unique class of scale-free transformation.
>
> Please also see item 1 regarding the scale-free condition in the common reply.
>
>
> We tried a direct neural-network learning-based fusion during the revision as suggested by Reviewer 3 and observed that the $\alpha$-fusion with $\alpha$ learning outperformed this direct method. Please see item 2 in the common response and Fig 6 and 7 in the revised paper.
>
>
>
>
> [Ref. Hardy et al., 1952] Hardy et al., "Inequalities," Cambridge Mathematical Library, 1952.
>
> [Ref Amari 2007], Amari, "Integration of stochastic models by minimizing $\alpha$-divergence," integration of stochastic models by minimizing $\alpha$-divergence, 2007.
>
> [Ref Amari 2016], Amari, "Information geometry and its applications," Springer, 2016.

---

### Official Review · AnonReviewer3 · 2020-10-30
**Contribution of the paper might be insufficient**

**Rating:** 5
**Confidence:** 4

**Review:**

Summary
The papers looks at the problem of using intrinsic rewards to help agent explore in sparse reward settings. The paper proposes combining multiple intrinsic rewards and proposes a meta-gradient based method to learning the fusion of these intrinsic rewards.


Strengths
1. Learning in a sparse rewards setting is a challenging and relevant problem to the community.
2. The paper is well written
3. The idea of learning to use multiple intrinsic rewards and using different combinations of them at different times seems to be helpful in the experiments

Weaknesses/Comments/Questions
1. My main concern is that the contribution of the paper might not be sufficient. The main contribution claimed by the paper to me seems like the idea of combining different intrinsic rewards and also learning the hyper-parameter that controls the fusion. In practice different intrinsic rewards are indeed used together and varied at different times. Also learning RL related hyperparameters using meta-gradient (Meta-gradient RL) is not new.
2. I don't think Zheng et al is really meant for sparse reward settings. So not sure if that's the right SOTA baseline to compare with. Zheng et al is not expected to learn intrinsic rewards that guide initial exploration as it needs extrinsic feedback to learn these intrinsic rewards, else they are just random.
3. It would be good if the paper evaluates on some domains that are naturally sparse reward setting instead of converting a dense reward setting to a sparse reward setting by accumulating and delaying rewards.
4. If the main contribution of the paper is on how to fuse, then simple methods of fusing like learning a lenear combination or just a single layer neural network over the different model prediction errors to output the intrinsic reward to use all seem reasonable. Comparison with those would be good.
5. The conditions that lead to the particular choice of fusion should be verified using ablation study to see if they indeed lead to the performance boost expected or if the performance comes from something else.
6. Clarification: Is it that all the experiments only fuse between two intrinsic rewards (prediction errors)? If the paper is about the importance of combining different prediction errors, it would be nice to have more than 2 and also some interestingly different model predictions and also analysis and comparison on what happens if we just use them separately.

Questions to authors.
1. Please respond to the comments above. Thanks.

---

> ### Author Response · Authors · 2020-11-24
> **Response to Reviewer 3**
>
> We thank you for your insightful comments. We added the responses by topic in detail.
>
> ### 1. Comparison to direct neural-network-based fusion functions and novelty of the proposed method
>
> We compared our fusion method with direct neural-network-based fusion and conducted visualization analysis in Appendix E on Pages 17-18 in the revised paper.
>
>
> In order to compare our fusion method to the fusion with neural network learning, we designed a neural network fusion function $f_\xi(x_1, \cdots, x_K)$ of $K=2$ inputs with (i) linear activation or (ii) nonlinear ($\tanh$) activation, as suggested by the reviewer. In both cases, $f_\xi$ has a single hidden layer of size $2K$.  Figure 6 in the revised paper shows that our fusion method outperforms the fusion with neural network learning using the same KLD model error input.
>
>
> It seems that the neural-network-based fusion function requires more complexity and more learning time. The proposed adaptive $\alpha$-fusion structure captures the fusion behavior only by using a single parameter $\alpha$, and the corresponding learning is efficient.
>
> Therefore, our fusion method is novel and efficient for sparse reward settings.
>
>
> ### 2. Additional baseline
>
> On page 6 in the main paper, the performance plots of additional baseline [Pathak et al., 2019] (suggested by Reviewer 1) are added in Figure 2 of the new version as yellow lines denoted as ‘Disagreement.’ It is observed that the proposed adaptive fusion method outperforms this baseline over the considered six environments.
>
> [Ref Pathak et al., 2019] Pathak et al., "Self-supervised exploration via disagreement," ICML 2019.
>
>
>
> ### 3. Regarding the baseline of [Zheng et al., 2018]
>
> The baseline of [Zheng et al., 2018] is useful for sparse reward settings. Although it needs some random initial exploration, the intrinsic reward is designed to maximize the cumulative sum of extrinsic reward and to help further exploration during the whole training process. This was verified by their results in [Zheng et al., 2018].
>
>
> [Ref Zheng et al., 2018] Zheng et al., "On learning intrinsic rewards for policy gradient methods," NeurIPS 2018.
>
>
> ### 4. Regarding a new environment
>
> We considered another sparse reward environment AntMazeSparse [Zhang et al., 2020]. In AntMazeSparse, an Ant robot in a U-shaped maze tries to achieve the target position called 'goal,' and the ant is rewarded by one only if it reaches the goal. We applied our method to AntMazeSparse and observed that our method did not perform well in  AntMazeSparse.  We will consider the extension of our method to the goal-conditioned RL setting as our future work.
>
>
> [Ref Zhang et al., 2020] Zhang et al., "Generating Adjacency-Constrained Subgoals in Hierarchical Reinforcement Learning," accepted to NeurIPS 2020.
>
>
>
>
> ### 5. Where does the performance improvement come from?
>
>
>
> The other setting except the intrinsic reward generation was the same for all the considered intrinsic reward generation methods. Hence, the performance difference is solely due to the intrinsic reward generation method.  The performance improvement of the proposed adaptive fusion comes from mainly two factors. The first one is that we use multiple prediction models, and hence we have diversity in the prediction values. The second one is that we optimally fuse these multiple prediction error values using the adaptive fusion rule, and the fusion itself is also learned. For detail, please see the ablation study in Section 4.2 of the paper.
>
>
> ### 6. The proposed method with mean-squared error (MSE)
>
>
> We compared the proposed method using KLD with the proposed method using MSE. In the MSE method, we trained ten deterministic models predicting the next state. These multiple models are differently initialized and independently trained with different batch data as widely done in other works [Nagabandi et al., 2017], [Kurutach et al., 2018]. Then, we calculated the MSE between the actual next state and the predicted next state as the error measure. It is observed that the proposed fusion with KLD outperforms the proposed fusion method using MSE (Fig. 8 in Appendix F, on page 19 of the new version).
>
> Since the KLD uses joint training among multiple probabilistic models, the KLD method can reflect the underlying dynamics more accurately. On the other hand, independently trained deterministic models were not diverse enough to effectively capture the underlying dynamics as compared to the joint training (so the gap between multiple MSE method and separated single MSE
> is small).
>
>
> [Ref Nagabandi et al., 2017] Nagabandi et al., "Neural network dynamics for model-based deep reinforcement learning with model-free fine-tuning," arXiv 2017.
>
> [Ref Kurutach et al., 2018] Kurutach et al., "Model-ensemble trust-region policy optimization," ICLR 2018.

---

### Author Response · Authors · 2020-11-24
**Common Response**

We thank all the reviewers for valuable and insightful feedback. We uploaded a revised version of our paper, with the modified parts shown in red.



### 1. The necessity of scale-free property (Condition 2)

Note that basically, the fusion function is some averaging function. Condition 2 is a proper condition for any reasonable averaging function [Hardy et al., 1952]. The scale-free property is also called the homogeneous property.

Suppose that the homogeneous property for the function function $f$ is not satisfied and assume the nonhomogeneous relationship $f(c x_1, c x_2) = c^\prime f(x_1, x_2)$,
where  $c^\prime ~(\ne c)$ is mapped for each scaling factor $c$ for the completeness of the scaling operation.


Now suppose that there exist two constants $c,c^\prime$ such that $c> 1$ and $0<c^\prime <1$ and  $ f(c x_1, c x_2) = c^\prime f(x_1, x_2)$
for two input $x_1$ and $x_2$. Then, the monotonicity is broken.  The two inputs $x_1$ and $x_2$ are increased as $cx_1$ and $c x_2$ but the corresponding output is reduced as $c^\prime f(x_1, x_2)$ as compared to the original output $f(x_1,x_2)$. This is not the situation that we want. Furthermore, by repeatedly applying $ f(c x_1, c x_2) = c^\prime f(x_1, x_2)$, we have
\begin{equation}
\lim_{n\rightarrow \infty} [f(c^nx_1,c^nx_2) = (c^\prime)^n f(x_1,x_2)]
\end{equation}
yielding
\begin{equation}
    f(\infty,\infty) = 0\cdot f(x_1,x_2) = 0.
\end{equation}
So, we have a contradiction.  In the case of $0 < c< 1$ and $c^\prime >1$, we have a similar contradiction:
\begin{equation}
    f(0,0) = \infty \cdot f(x_1,x_2) = \infty.
\end{equation}

Now assume that there exist $c, c^\prime >1$ and $c \ne c^\prime$. Set two inputs as $x_1=x_2=x$. Then, we have
\begin{eqnarray}
g(x) &\stackrel{\triangle}{=}& f(x,x), \\
g(cx) &=& f(cx,cx) = c^\prime f(x,x)= c^\prime g(x).
\end{eqnarray}
By repeating the iteration, we have
\begin{equation}
    g(c^n x) = (c^\prime)^n g(x).
\end{equation}



In this equation, the input to the function $g(x)$ is exponentially increasing as $c^n$ and the output increases exponentially as $(c^\prime)^n$. Such function $g(x)$ is uniquely given by the form
\begin{equation}
g(x) \sim (c^\prime)^{\log_c x},
\end{equation}
where $\sim$ means the scaling equivalence.
However, for a different pair $\tilde{c}$ and $\tilde{c}^\prime$, we also require
\begin{equation}
g(x) \sim (\tilde{c}^\prime)^{\log_{\tilde{c}} x}.
\end{equation}
The two functions above  cannot be the same in general.
Furthermore, $g(x)$ should be the same for all pairs $(c,c^\prime)$. This cannot be satisfied in general.
So, we have an indiscrepancy in the nonhomogeneous case. A similar situation happens for $c,c^\prime < 1$. However, note that if we have $c=c^\prime$ for all scaling factor $c$, then the two functions $(c^\prime)^{\log_c x}$ and $(\tilde{c}^\prime)^{\log_{\tilde{c}} x}$ are consistent. In this case, we have
\begin{equation}
    g(x) \sim (c)^{\log_c x} =x,
\end{equation}
and this makes sense because if the input values are all the same, the output should be the same as the input. Please note that the generalized mean or $\alpha$-mean exactly satisfies the scaling behavior.


[Ref. Hardy et al., 1952] Hardy et al., "Inequalities," Cambridge Mathematical Library, 1952.



### 2. Comparison to direct neural-network-based fusion functions

As suggested by Reviewer 3, we compared our fusion method with direct neural-network-based fusion and conducted visualization analysis in Appendix E on Pages 17-18.


In order to compare our fusion method to the fusion with neural network learning, we designed a neural network fusion function $f_\xi(x_1, \cdots, x_K)$ of $K=2$ inputs with (i) linear activation or (ii) nonlinear ($\tanh$) activation. In both cases, $f_\xi$ has a single hidden layer of size $2K$.  Fig. 6 in the revised paper shows that our method outperforms the fusion with neural network learning using the same KLD model error input.


It seems that the neural-network-based fusion function requires more complexity and more learning time. The proposed adaptive $\alpha$-fusion structure captures the fusion behavior only by using a single parameter $\alpha$, and the corresponding learning is efficient.

---

### Author Response · Authors · 2020-11-24
**Common Response Continued**


### 3. The proposed method with mean-squared error (MSE)


We compared the proposed method using KLD with the proposed method using MSE. In the MSE method, we trained ten deterministic models predicting the next state. These multiple models are differently initialized and independently trained with different batch data as widely done in other works [Nagabandi et al., 2017], [Kurutach et al., 2018]. Then, we calculated the MSE between the actual next state and the predicted next state as the error measure. It is observed that the proposed fusion with KLD outperforms the proposed fusion method using MSE (Fig. 8 in Appendix F, on page 19 of the new version).

Since the KLD uses joint training among multiple probabilistic models, the KLD method can reflect the underlying dynamics more accurately. On the other hand, independently trained deterministic models were not diverse enough to effectively capture the underlying dynamics compared to the joint training.


[Ref Nagabandi et al., 2017] Nagabandi et al., "Neural network dynamics for model-based deep reinforcement learning with model-free fine-tuning," arXiv 2017.

[Ref Kurutach et al., 2018] Kurutach et al., "Model-ensemble trust-region policy optimization," ICLR 2018.




### 4. Additional baseline

On page 6 in the main paper, the performance plots of additional baseline [Pathak et al., 2019] (suggested by Reviewer 1) are added in Fig. 2 of the new version as yellow lines denoted as ‘Disagreement.’ It is observed that the proposed adaptive fusion method outperforms this baseline over the considered six environments.

[Ref Pathak et al., 2019] Pathak et al., "Self-supervised exploration via disagreement," ICML 2019.



### 5. Clarification

Footnote 1  at the bottom of page 4 was clarified.

---

### Author Response · Authors · 2020-11-25
**The necessity of Condition 2 is added in the revised version**

We uploaded the new revised version of the paper, including:

1. The necessity of Condition 2 was added in Appendix G on page 20.

2. Section 2.2 in 'Related Work' was moved to Appendix H on page 21 (suggested by Reviewer 4).

3. References were updated to the published version from the arxiv version when the former is available.

---

### Decision · Program_Chairs · 2021-01-07
**Final Decision**

**Decision:**

Reject

**Comment:**

The paper extends previous work on intrinsic reward design based on curiosity or surprise toward multiple intrinsic rewards based multiple model predictions and fuse the reward using meta-gradient optimization.  While most reviewers find the paper clearly written, several reviewers do bring up the concern on limited contribution of the work on top of existing ones. Reviewers also would like to see experiments conducted in environment with sparse reward rather than the delayed reward setting constructed from dense reward environments. More ablation studies on the different design choices will also be helpful.